

# Nomograms predict survival outcome of Klatskin tumors patients

Feng Qi[1,*], Bin Zhou[2,*] and Jinglin Xia[1]

[1] Liver Cancer Institute, Zhongshan Hospital, Fudan University, Shanghai, China
[2] Department of Hepatic Surgery VI, Eastern Hepatobiliary Surgery Hospital, Second Military Medical University, Shanghai, China
[*] These authors contributed equally to this work.

## ABSTRACT

**Objective**. Klatskin tumors are rare, malignant tumors of the biliary system with a poor prognosis for patient survival. The current understanding of these tumors is limited to a small number of case reports or case series; therefore, we examined prognostic factors of this disease.

**Methods**. A population cohort study was conducted in patients selected from the Surveillance, Epidemiology, and End Results (SEER) database with a Klatskin tumor that was histologically diagnosed between 2004 to 2014. Propensity-matching (PSM) analysis was performed to determine the overall survival (OS) among those with a Klatskin tumor (KCC), intrahepatic cholangiocarcinoma (ICCA), or hepatocellular carcinoma (HCC). The nomogram was based on 317 eligible Klatskin tumor patients and its predictive accuracy and discriminatory ability were determined using the concordance index (C-index).

**Results**. Kaplan-Meier analysis showed that patients with Klatskin tumors had significantly worse overall survival rates (1-year OS = 26.2%, 2-year OS = 10.7%, 3-year OS = 3.4%) than those with intrahepatic cholangiocarcinoma (1-year OS = 62.2%, 2-year OS = 36.4%, 3-year OS = 19.1%, $p < 0.001$) or hepatocellular carcinoma (1-year OS = 72.4%, 2-year OS = 48.5%, 3-year OS = 36.2%, $p < 0.001$). A poor prognosis was also significantly associated with older age, higher grade, SEER historic stage, and lymph node metastasis. Local destruction of the tumor (HR = 0.635, 95% CI [0.421–0.956], $p = 0.03$) and surgery (HR = 0.434, 95% [CI 0.328–0.574], $p < 0.001$) were independent protective factors. Multivariate Cox analysis showed that older age, SEER historic stage, and lymph node metastases (HR = 1.468, 95% CI [1.008–2.139], $p = 0.046$) were independent prognostic factors of poor survival rates in Klatskin tumor patients, while cancer-directed surgery was an independent protective factor (HR = 0.555, 95% CI [0.316–0.977], $p = 0.041$). The prognostic and protective factors were included in the nomogram (C-index for survival = 0.651; 95% CI [0.607–0.695]).

**Conclusions**. The Klatskin tumor group had poorer rates of OS and cancer-specific survival than the ICCA and HCC groups. Early detection and diagnosis were associated with a higher rate of OS in Klatskin tumor patients.

Corresponding author
Jinglin Xia, xiajinglin@fudan.edu.cn

## INTRODUCTION

Cholangiocarcinoma (CCA) is a heterogeneous, malignant tumor with an extremely poor prognosis for survival. It is the second most common primary hepatic malignancy and makes up 3% of all gastrointestinal tumors (*Blechacz & Gores, 2008*). Studies conducted in North America, Europe, Asia, and Australia have shown a steady increase in the incidence of CCA over the past few decades (*Patel, 2001*). The cause of this increase is unclear but may be attributed to several risk factors, including primary sclerosing cholangitis, ulcerative colitis, cirrhosis, hepatitis B, infection by certain liver flukes, and some congenital liver malformations (*Razumilava & Gores, 2014*). CCA is thought to develop in a histologically similar manner to colon cancer, with a series of stages from early hyperplasia and metaplasia to dysplasia and, ultimately, to the development of cancer (*Sirica, 2005*). CCA can develop in any area of the bile duct, including in the bile ducts within the liver (intrahepatic), outside of the liver (extrahepatic), and in the perihilar region (*DeOliveira et al., 2007*).

In 1965, Klatskin identified 13 cases of adenocarcinoma in the hepatic duct bifurcation and named this tumor type the Klatskin tumor, or hilar cholangiocarcinoma (HCCA) (*Okuda et al., 1977*). Klatskin tumors were then further divided into two types due to their poor definition in the extrahepatic and intrahepatic bile ducts, known as extrahepatic hilar cholangiocarcinoma (EHC) and intrahepatic hilar cholangiocarcinoma (IHC) (*Okuda et al., 1977*), respectively; they have similar histological characteristics based on the International Classification of Disease-Oncology (ICD-O) and *Ebata et al.*'s *2009* report. Previous studies have primarily focused on case reports or series based on the anatomical location of this disease while cohort studies performed have only examined the annual incidence and treatment of this disease (*Sharma & Yadav, 2018*). The prognostic factors of Klatskin tumors have not yet been compared with those of ICCA and HCC.

We retroactively analyzed data taken from the Surveillance, Epidemiology, and End Results (SEER) database for patients diagnosed with a Klatskin tumor or ICCA. HCC was used as a point of reference to compare the clinical features, prognostic factors, and overall survival (OS) between Klatskin tumor and ICCA patients in order to further explore the factors that influence OS.

## MATERIAL AND METHODS

### Patient data

Stat version 8.2.1 of SEER was used to download data from all patients with a diagnosis of liver cancer and cholangiocarcinoma from 2005 to 2015, according to the ICD-0-3/WHO 2008 guidelines. The main inclusion criteria were as follows: age 0 to 104 years, liver cancer with cholangiocarcinoma as the main malignant cancer diagnosis, pathological types of Klatskin tumors (based on ICD-O-3 8162/3), ICCA (based on ICD-O-3 8160/3), and HCC (based on ICD-O-3 8170/3). We also examined the gender, ethnicity, lymph node metastases, histological grades I to IV, definite AJCC TNM stages, and type of therapeutic strategy. We excluded patients with an unclear T stage record, and unknown survival time, diagnostic confirmation, or surgery strategy. Patients were included based on a diagnosis of liver cancer and cholangiocarcinoma before 2014 to ensure sufficient time for follow-up
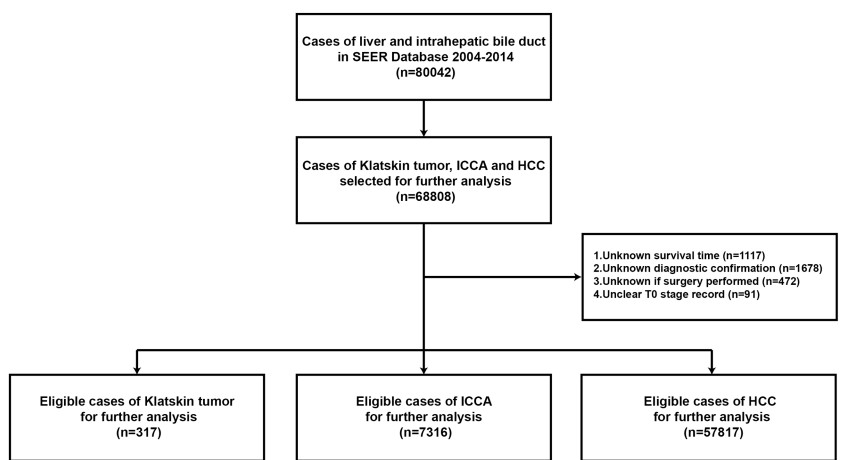

**Figure 1   Flowchart of the enrolled patients in the study according to inclusion and exclusion crite-rion.**

(Fig. 1). The study was in compliance with the ethics statement of Zhongshan Hospital, Fudan University.

## Statistical analysis

Normalized continuous variables with a homogeneity of variance were compared using the t test; the Mann–Whitney U test was performed for all other variable calculations. Multiple groups were compared using ANOVA and Tukey's post hoc test. The chi-square test was applied to the categorical data. Patients were divided into three groups according to pathology: Klatskin tumor group, the ICCA group, and the HCC group. The demographic and clinical characteristics of the three groups were compared using the chi-square test. Continuous variables were presented as mean $\pm$ SE or median (minimum, maximum), and the categorical variables were calculated as a percentage. PSM analyses were performed based upon age, race, gender, grade, historic stage, AJCC 6th stage, tumor size, lymph node status, surgical method, alpha-fetoprotein (AFP) level, and fibrosis score at a 1:1 ratio to adjust for the differences among the Klatskin tumor, ICCA, and HCC groups. The Kaplan–Meier method was used to plot the survival curve. The unadjusted OS rate of the different histological subtypes was adjusted using the log-rank test. OS was defined from the date of diagnosis to the date of death or final follow-up. Cox proportional hazard regression models were used to evaluate prognostic factors and HR was used as the 95% confidence interval. Diagnostic age, summary stage, lymph node metastases, and the treatment of the primary tumor were included in the survival analysis. The above statistical data were analyzed using the SPSS version 22.0 software package (IBM SPSS Statistics, Chicago, IL, US) and $p < 0.05$ was considered statistically significant.
# RESULTS

## Demographic, tumor, and treatment characteristics of patients with Klatskin tumors, ICCA, or HCC

A total of 65,450 patients met the inclusion criteria for our study, including 317 Klatskin tumor patients, 7,316 ICCA patients, and 57,817 HCC patients. The demographic and clinical characteristics of all patients are summarized in Table 1. The composition of the three groups differed significantly by age, race, gender, degree of differentiation, AJCC stage, surgical method, alpha fetoprotein (AFP), and fibrosis score. Klatskin tumors occurred more frequently in older patients ($72.78 \pm 13.29, 66.76 \pm 12.99$ vs $63.62 \pm 11.48, p < 0.001$) and among white men, which is similar to the demographics of the ICCA and HCC groups (79.5%, 78.3% vs 68.3%, $p < 0.001$). The localized and regional historic stages for patients with Klatskin tumors was 31.2% and 31.9%, 26.8% and 30.2% for the ICCA group, and 49.7% and 27.9% for the HCC group, respectively ($p < 0.001$). We lacked sufficient data to generate statistically significant results when running comparison analyses of the treatment method. The overall incidence rate of Klatskin tumors decreased between 2004 and 2014 ($r = -0.94, p < 0.001$) while the rates of ICCA ($r = 0.89, p < 0.001$) and HCC ($r = 0.73, p = 0.011$) increased (Fig. 2).

## Survival and prognostic factors for Klatskin tumor, ICCA, and HCC patients

Kaplan–Meier analysis was used to evaluate OS and cancer-specific survival (Fig. 2). Klatskin tumor patients had worse survival rates than ICCA and HCC patients (median OS: 5 months vs 9 months and 14 months, $p < 0.001$; median cancer-specific survival: 10 months vs 13 months and 23 months, $p < 0.001$). The 1-, 2- and 3-year OS rates of Klatskin tumor, ICCA, and HCC patients were 26.2% versus 62.2% and 72.4%, 10.7% versus 36.4% and 48.5%, and 3.4% versus 19.1% and 36.2%, respectively. The X-tile program was used to divide all patients into two groups according to age in order to more accurately determine the prognosis. Further stratification studies showed that age at diagnosis, race, gender, tumor differentiation, SEER historic stage, tumor size, lymph node status, surgical intervention (including local tumor destruction and surgery), AFP level, and fibrosis score had a significant impact on the OS of Klatskin tumor, ICCA, and HCC patients ($p < 0.05$) (Fig. 3).

The results of the Kaplan–Meier analysis of OS were further analyzed using univariate and multivariate Cox regression models. According to the univariate factor analysis, older age, worse pathological grade, larger tumor size, lymph node metastases, lower AFP level, and late TNM and SEER stages were significantly related to a worse prognosis ($p < 0.05$) (Table 2). In contrast, surgical intervention and histologic type were low risk factors ($p < 0.001$). Adjusting the variables did not impact the results of the multivariate analysis, except in regard to lymph node metastases ($p = 0.089$).

PSM analysis was performed to adjust for the unmatching cohort and a total of 317 Klatskin tumor patients were matched with 317 ICCA and 317 HCC patients (1:1:1) (Table 3). In the matching cohorts, the Klatskin tumor group had worse 1-, 2-, 3-year OS rates than the ICCA and HCC groups (1-year OS: 28.4% versus 71.9% and 85.6%;

**Table 1  Characteristics of patients with Klatskin tumors, ICCA and HCC in the SEER database, 2004–2014.**

| Characteristics | Histologic type | | | | |
|---|---|---|---|---|---|
| | Klatskin tumor | ICCA | HCC | Total | *p*-value |
| **Number** | 317 | 7316 | 57817 | 65450 | |
| **Age (years)** | 72.78 ± 13.29 | 66.76 ± 12.99 | 63.62 ± 11.48 | | <0.001 |
| **Marital status** | | | | | |
| Married | 166(52.4%) | 4163(56.9%) | 29315(50.7%) | 33644(51.4%) | <0.001 |
| Not married[a] | 137(43.2%) | 2849(38.9%) | 25778(44.6%) | 28764(43.9%) | |
| Unknown | 14(4.4%) | 304(4.2%) | 2724(4.7%) | 3042(4.7%) | |
| **Race** | | | | | |
| White | 252(79.5%) | 5729(78.3%) | 39500(68.3%) | 45481(69.5%) | <0.001 |
| Black | 21(6.6%) | 589(8.0%) | 7832(13.5%) | 8442(12.9%) | |
| Other[b] | 44(13.9%) | 987(13.5%) | 10294(17.8%) | 11325(17.3%) | |
| Unknown | 0(0%) | 11(0.2%) | 191(0.4%) | 202(0.3%) | |
| **Gender** | | | | | |
| Female | 141(44.5%) | 3669(50.2%) | 13374(23.1%) | 17184(26.3%) | <0.001 |
| Male | 176(55.5%) | 3647(49.8%) | 44443(76.9%) | 48266(73.7%) | |
| **Grade** | | | | | |
| Well differentiated | 9(2.8%) | 319(4.4%) | 6819(11.8%) | 7147(10.9%) | <0.001 |
| Moderate | 15(4.7%) | 1435(19.6%) | 9140(15.8%) | 10590(16.2%) | |
| Poor | 16(5.0%) | 1268(17.3%) | 4533(7.8%) | 5817(8.9%) | |
| Anaplastic | 1(0.3%) | 39(0.5%) | 398(0.7%) | 438(0.7%) | |
| Unknown | 276(87.2%) | 4255(58.2%) | 36927(63.9%) | 41458(63.3%) | |
| **SEER historic stage** | | | | | |
| Localized | 99(31.2%) | 1963(26.8%) | 28758(49.7%) | 30820(47.1%) | <0.001 |
| Regional | 101(31.9%) | 2208(30.2%) | 16106(27.9%) | 18415(28.1%) | |
| Distant | 51(16.1%) | 2402(32.8%) | 8699(15.0%) | 11152(17.0%) | |
| Unknown | 66(20.8%) | 743(10.2%) | 4254(7.4%) | 5063(7.8%) | |
| **AJCC stage** | | | | | |
| I | 76(24.0%) | 1388(19.0%) | 19005(32.9%) | 20469(31.3%) | <0.001 |
| II | 14(4.4%) | 476(6.5%) | 9865(17.1%) | 10355(15.8%) | |
| III | 72(22.7%) | 1622(22.2%) | 11569(20.0%) | 13263(20.3%) | |
| IV | 52(16.4%) | 2405(32.9%) | 8331(14.4%) | 10788(16.5%) | |
| Unknown | 103(32.5%) | 1425(19.4%) | 9047(15.6%) | 10575(16.1%) | |
| **T category** | | | | | |
| T1 | 118(37.2%) | 2216(30.3%) | 22488(38.9%) | 24822(37.9%) | <0.001 |
| T2 | 28(8.8%) | 826(11.3%) | 11855(20.5%) | 12709(19.4%) | |
| T3 | 37(11.7%) | 1481(20.2%) | 12808(22.2%) | 14326(21.9%) | |
| T4 | 35(11.0%) | 812(11.1%) | 2201(3.8%) | 3048(4.7%) | |
| Tx | 99(31.3%) | 1981(27.1%) | 8465(14.6%) | 10545(16.1%) | |
| **N category** | | | | | |
| N0 | 184(58.0%) | 4441(60.7%) | 45479(78.7%) | 50104(76.6%) | <0.001 |
| N1 | 56(17.7%) | 1396(19.1%) | 3752(6.5%) | 5204(8.0%) | |

*(continued on next page)*

**Table 1** (*continued*)

| Characteristics | Histologic type | | | | |
| --- | --- | --- | --- | --- | --- |
| | Klatskin tumor | ICCA | HCC | Total | *p*-value |
| Nx | 77(24.3%) | 1479(20.2%) | 8586(14.8%) | 10142(15.4%) | |
| **Local treatment of the primary tumor** | | | | | |
| None | 293(92.4%) | 5721(78.2%) | 43690(75.6%) | 49704(75.9%) | <0.001 |
| Local destruction | 0(0.0%) | 154(2.1%) | 5936(10.3%) | 6090(9.3%) | |
| Surgery | 24(7.6%) | 1441(19.7%) | 8191(14.1%) | 9656(14.8%) | |
| **Alpha fetoprotein** | | | | | |
| Elevated | 22(6.9%) | 967(13.2%) | 33014(57.1%) | 34003(52.0%) | <0.001 |
| Normal | 62(19.6%) | 2297(31.4%) | 10483(18.1%) | 12842(19.6%) | |
| Unknown | 233(73.5%) | 4052(55.4%) | 14320(24.8%) | 18605(28.4%) | |
| **Fibrosis score** | | | | | |
| 0-4 | 14(4.4%) | 450(6.2%) | 2781(4.8%) | 3245(5.0%) | <0.001 |
| 5-6 | 2 (0.6%) | 263(3.6%) | 11562(20.0%) | 11827(18.0%) | |
| Unknown | 301(95.0%) | 6603(90.2%) | 43474(75.2%) | 50378(77.0%) | |

**Notes.**
[a]Including divorced, separated, single (never married), unmarried or having a domestic partner and widowed.
[b]Including American Indian/Alaskan Native and Asian/Pacific Islander.
AJCC, American Joint Committee on Cancer; ICCA, Intrahepatic Cholangiocarcinoma; HCC, hepatocellular carcinoma.

2-year OS: 13.3% versus 52.5% and 73.0%; 3-year OS: 5.2% versus 38.1% and 66.1%). The median OS in the Klatskin tumor group was 5 months compared with 26 and 68 months in the ICCA and HCC groups, respectively. The median cancer-specific survival in the Klatskin tumor group was 10 months compared with 34 months in the ICCA group ($p < 0.001$) (Fig. 4). Kaplan–Meier analysis and univariate Cox regression was used for further stratification studies and the results were similar to those of the un-matched cohorts (Fig. 5). Based on multivariable analysis, the prognostic factors for OS were older age (HR 1.736, 95% CI [1.396–2.157], $p < 0.001$), lymph node metastasis (HR 1.406, 95% CI [1.093–1.808], $p = 0.008$), surgical intervention, lower AFP level (HR 0.757, 95% CI [0.587–0.976], $p = 0.032$) and histologic type (Table 4).

## Klatskin tumor patient survival and prognostic factors

317 patients were identified with Klatskin tumors from 2004 to 2014 and a Kaplan–Meier analysis was performed based on patient characteristics (Table 1). Kaplan–Meier survival curves and log-rank analysis showed that a poorer prognosis was associated with older age (median OS: 3 months versus 7 months, $p < 0.001$), distant stage (median OS: localized 7 months, regional 6 months versus distant 2 months, $p < 0.001$), no surgery (median OS: 5 months vs 17 months, $p < 0.001$), and lymph node metastases (median OS: 3 months vs 7 months, $p = 0.016$) (Fig. 6).

Multivariate Cox analysis of the 317 Klatskin tumor patients showed that older age (HR 1.725, 95% CI [1.324–2.249], $p < 0.001$), late SEER historic stage (HR 3.594, 95% CI [1.251–10.326], $p = 0.018$), and lymph node metastases (HR=1.468, 95% CI [1.008–2.139], $p = 0.046$) were independent prognostic factors for worse survival. Conversely, cancer-directed surgery was an independent protective factor that decreased the risk of

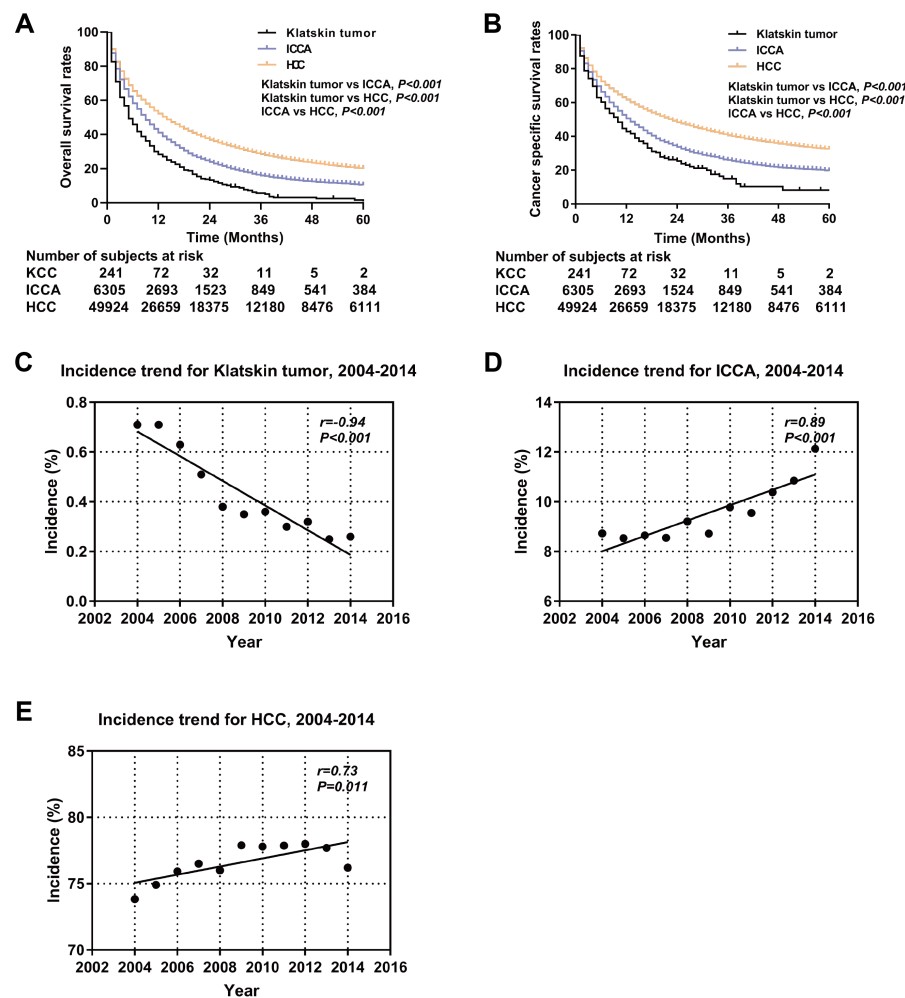

**Figure 2** **Kaplan-Meier curve of overall survival and the incidence trend.** Overall survival (A) and cancer specific survival (B) according to histologic type of Klatskin tumors, ICCA or HCC before propensity score matching. The incidence trend for Klatskin tumor (C), ICCA (D) and HCC (E).

death by 44.5% for Klatskin tumor patients (HR=0.555, 95% CI [0.316–0.977], $p = 0.041$) (Table 5).

## Prognostic nomogram for Klatskin tumor

The prognostic nomogram was used to integrate all of the significant independent factors for OS in the Klatskin tumor group (Fig. 7). The patients were divided into three groups according to age using the X-tile program. The optimal cut-off points were at 71 and 82 years of age. A prognostic nomogram was established to more accurately determine the survival of Klatskin tumor patients using all of the significant independent variables based on multivariate Cox analysis (Fig. 7A). The C-index for OS prediction was 0.651 (95% CI [0.607–0.695]). The calibration plot for the probability of survival at 1, 2, and 3 years showed an optimal agreement between the prediction by nomogram and actual

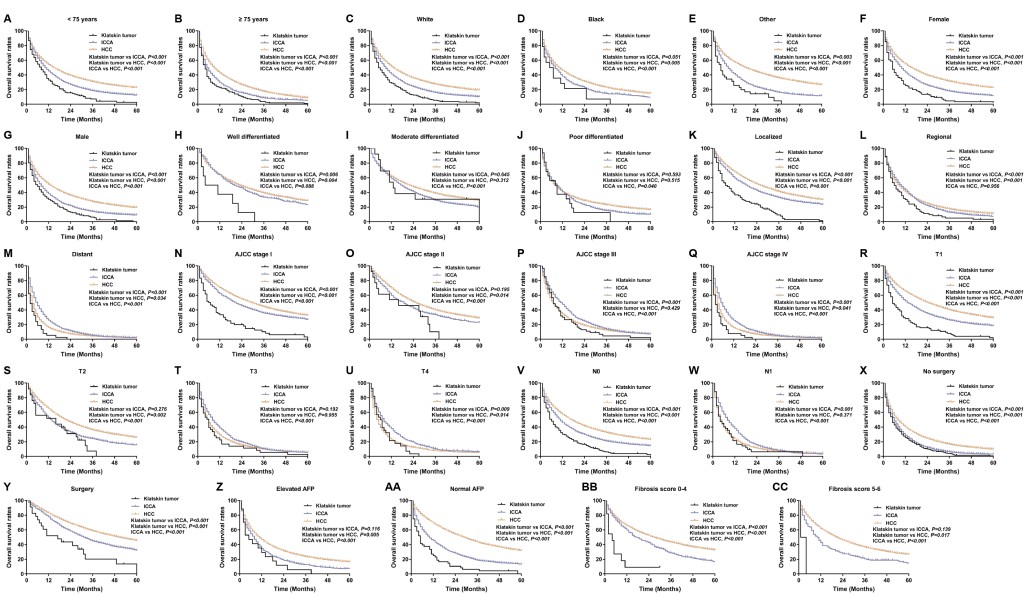

**Figure 3** Kaplan–Meier curve of overall survival in Klatskin tumor, ICCA and HCC patients before propensity score matching stratified by (A, B) age at diagnosis; (C-E) race; (F, G) gender; (H-J) primary tumor differential grade; (K-M) SEER historic stage; (N-Q) AJCC stage; (R-U) primary tumor size; (V, W) lymph node status; (X, Y) surgery for primary tumor; (Z, AA) AFP level; (BB, CC) fibrosis score.

observation (Figs. 7B-D). The ROC curve was used to predict the accuracy of the model and the analysis was in agreement with the nomogram with a value of 0.884 in the area under the curve (AUC) ($p < 0.001$, Fig. 7E).

## DISCUSSION

Klatskin tumors are rare but their incidence has been on the rise since their identification in 1965 (*Okuda et al., 1977*). However, the clinicopathological features and outcomes of this disease remain unclear. In our study, we evaluated the clinicopathological features and prognostic factors of Klatskin tumors using data from the SEER database from 2004 to 2014. We found the prevalence of Klatskin tumors from this database to be very low when compared with intrahepatic cholangiocarcinoma (ICCA) (317 versus 7,316 cases, respectively).

The mean age at diagnosis was 73, which is consistent with the report by *Sharma & Yadav (2018)*. White patients were predominant in this cohort (75.0%), which was expected given the overall racial distribution of the Western population. Of the 317 cases, more males were affected than females, but this difference was small (176:141). Most Klatskin tumor patients did not undergo surgery or local destruction of the tumor. Patients with regional and localized tumors in the early stages of the disease accounted for 63.1% of the Klatskin tumor cohort.

Klatskin tumors were found to have a worse prognosis than ICCA or hepatocellular carcinoma and were more clinically aggressive with a worse prognosis than HCC. The median OS for patients with CCA of the pancreas was nearly half that of patients with

**Table 2** Univariate and multivariate Cox proportional hazard analyses of the association of clinical characteristics with overall survival rates in patients with Klatskin tumors, ICCA and HCC.

| Variance | Univariate HR (95% CI) | *P* value | Multivariate HR (95% CI) | *P* value |
|---|---|---|---|---|
| **Age** | | | | |
| <75 years | 1 | | 1 | |
| ≥75 years | 1.519(1.489–1.550) | <0.001 | 1.414(1.385–1.443) | <0.001 |
| **Race** | | | | |
| White | 1 | | 1 | |
| Black | 1.127(1.099–1.156) | <0.001 | 1.080(1.053–1.108) | <0.001 |
| Other | 0.820(0.801–0.839) | <0.001 | 0.830(0.811–0.850) | <0.001 |
| **Gender** | | | | |
| Female | 1 | | 1 | |
| Male | 1.051(1.031–1.071) | <0.001 | 1.088(1.067–1.110) | <0.001 |
| **Grade** | | | | |
| Well differentiated | 1 | | 1 | |
| Moderate | 1.029(0.993–1.065) | 0.113 | 1.159(1.119–1.200) | <0.001 |
| Poor | 1.694(1.630–1.760) | <0.001 | 1.496(1.438–1.556) | <0.001 |
| Anaplastic | 1.763(1.589–1.956) | <0.001 | 1.561(1.407–1.733) | <0.001 |
| **SEER historic stage** | | | | |
| Localized | 1 | | 1 | |
| Regional | 1.940(1.901–1.980) | <0.001 | 1.279(1.245–1.314) | <0.001 |
| Distant | 3.456(3.374–3.540) | <0.001 | 1.527(1.396–1.671) | <0.001 |
| **AJCC stage** | | | | |
| I | 1 | | 1 | |
| II | 1.019(0.991–1.048) | 0.181 | 1.000(0.942–1.062) | 0.992 |
| III | 2.340(2.284–2.398) | <0.001 | 1.296(1.235–1.360) | <0.001 |
| IV | 3.809(3.711–3.910) | <0.001 | 1.545(1.404–1.700) | <0.001 |
| **Stage T** | | | | |
| T1 | 1 | | 1 | |
| T2 | 1.010(0.985–1.035) | 0.439 | 0.951(0.902–1.002) | 0.059 |
| T3 | 2.271(2.221–2.323) | <0.001 | 1.249(1.200–1.300) | <0.001 |
| T4 | 2.462(2.366–2.561) | <0.001 | 1.221(1.159–1.285) | <0.001 |
| **Stage N** | | | | |
| N0 | 1 | | 1 | |
| N1 | 1.996(1.937–2.056) | <0.001 | 1.029(0.996–1.063) | 0.089 |
| **Local treatment of the primary tumor** | | | | |
| None | 1 | | 1 | |
| Local destruction | 0.387(0.375–0.400) | <0.001 | 0.521(0.504–0.538) | <0.001 |
| Surgery | 0.237(0.230–0.244) | <0.001 | 0.311(0.301–0.322) | <0.001 |
| **Alpha fetoprotein** | | | | |
| Elevated | 1 | | 1 | |
| Normal | 0.642(0.627–0.657) | <0.001 | 0.731(0.714–0.749) | <0.001 |
| **Fibrosis score** | | | | |
| 0-4 | 1 | | 1 | |

**Table 2** (*continued*)

| Variance | Univariate HR (95% CI) | *P* value | Multivariate HR (95% CI) | *P* value |
|---|---|---|---|---|
| 5-6 | 1.114(1.065–1.167) | <0.001 | 0.962(0.918–1.007) | 0.100 |
| **Histologic type** | | | | |
| Klatskin tumor | 1 | | 1 | |
| ICCA | 0.674(0.601–0.756) | <0.001 | 0.726(0.647–0.814) | <0.001 |
| HCC | 0.517(0.462–0.578) | <0.001 | 0.704(0.628–0.788) | <0.001 |

**Notes.**

CI, confidence interval; ICCA, Intrahepatic Cholangiocarcinoma; HCC, hepatocellular carcinoma.

HCC (*Bridgewater et al., 2014*; *Sapisochin et al., 2014*). The prognosis of Klatskin tumors was closely related to age, lymph node status, summary stage, and local treatment of the primary tumor according to multivariate regression analysis. Overall survival (OS) was influenced by summary stage, lymph node metastases, and surgery.

The method described by the American Joint Committee on Cancer (AJCC) is typically used in a clinical setting to determine a prognosis, however, the SEER historic stage has consistent definitions and disease progression measures that clearly demonstrate the poor prognosis of Klatskin tumor patients (*Edge & Compton, 2010*). Among the 317 Klatskin tumor patients, 103 were not categorized by AJCC stage. Thus, we adjusted the SEER historic stage based on other variables and included these data in the multivariate Cox analysis. The results showed that the SEER historic stage is likely associated with poor prognosis. Klatskin tumors have different etiologies and biological features than ICCA and HCC but the treatment is similar (*Deoliveira et al., 2011*; *Mansour et al., 2015*). Previous studies have shown a correlation between radical surgical excision and tumor characteristics and stage (*Scurtu et al., 2017*; *Ito et al., 2009*). *Juntermanns et al. (2016)* reported an improved long-term survival rate after the tumor and caudate lobe were completely resected. The results of our study were consistent with previous reports that patients who underwent surgery had much better rates of survival than those who did not. The local destruction of tumors was performed more frequently in HCC patients and was a confounding factor that impacted the difference in prognosis between the HCC and Klatskin tumor groups. This confounding factor caused a shorter OS in Klatskin tumor patients. Patients were more likely to be treated with surgery when the cancer was regional and localized and when the disease was at an early TNM stage. *Molina et al. (2015)* also reported high mortality rates in Klaskin tumor patients related to vascular metastases and lymph node involvement. Lymph node metastases were an independent prognostic protective factor for Klatskin tumor patients in our cohort. The median OS of Klatskin tumor patients with no lymph node metastases was relatively longer (7 months). Nomograms are thought to be more accurate than the conventional staging systems for predicting prognosis in some cancers (*Touijer & Scardino, 2009*; *Wang et al., 2013*). We constructed a prognostic nomogram, which performed well in predicting survival in Klatskin tumor patients as indicated by the C-index (0.651) and the calibration curve.

There are several hypotheses regarding the etiology of Klatskin tumors although its origins are still unknown. The most likely causes are primary sclerosing cholangitis, ulcerative colitis, cirrhosis, hepatitis B, infection with certain liver flukes, and certain

**Table 3  Characteristics of patients with Klatskin tumors, ICCA and HCC after propensity score matching in the SEER database, 2004–2014.**

| Characteristics | Histologic type | | | | |
| --- | --- | --- | --- | --- | --- |
| | Klatskin tumor | ICCA | HCC | Total | p-value |
| **Number** | 317 | 317 | 317 | 951 | |
| **Age (years)** | 72.78 ± 13.29 | 59.57 ± 13.14 | 58.73 ± 11.36 | | <0.001 |
| **Marital status** | | | | | |
| Married | 166(52.4%) | 213(67.2%) | 144(45.4%) | 523(55.0%) | <0.001 |
| Not married[a] | 137(43.2%) | 97(30.6%) | 153(48.3%) | 387(40.7%) | |
| Unknown | 14(4.4%) | 7(2.2%) | 20(6.3%) | 41(4.3%) | |
| **Race** | | | | | |
| White | 252(79.5%) | 225(71.0%) | 88(27.8%) | 565(59.4%) | <0.001 |
| Black | 21(6.6%) | 37(11.7%) | 63(19.9%) | 121(12.7%) | |
| Other[b] | 44(13.9%) | 55(17.3%) | 157(49.5%) | 256(26.9%) | |
| Unknown | 0(0%) | 0(0%) | 9(2.8%) | 9(1.0%) | |
| **Gender** | | | | | |
| Female | 141(44.5%) | 222(70.0%) | 3(0.9%) | 366(38.5%) | <0.001 |
| Male | 176(55.5%) | 95(30.0%) | 314(99.1%) | 585(61.5%) | |
| **Grade** | | | | | |
| Well differentiated | 9(2.8%) | 98(30.9%) | 157(49.5%) | 264(27.8%) | <0.001 |
| Moderate | 15(4.7%) | 187(59.0%) | 144(45.5%) | 346(36.4%) | |
| Poor | 16(5.0%) | 32(10.1%) | 16(5.0%) | 64(6.7%) | |
| Anaplastic | 1(0.3%) | 0(0.0%) | 0(0.0%) | 1(0.1%) | |
| Unknown | 276(87.2%) | 0(0.0%) | 0(0.0%) | 276(29.0%) | |
| **SEER historic stage** | | | | | |
| Localized | 99(31.2%) | 125(39.4%) | 266(83.9%) | 490(51.5%) | <0.001 |
| Regional | 101(31.9%) | 101(31.9%) | 47(14.8%) | 249(26.2%) | |
| Distant | 51(16.1%) | 76(24.0%) | 4(1.3%) | 131(13.8%) | |
| Unknown | 66(20.8%) | 15(4.7%) | 0(0.0%) | 81(8.5%) | |
| **AJCC stage** | | | | | |
| I | 76(24.0%) | 75(23.7%) | 189(59.6%) | 340(35.8%) | <0.001 |
| II | 14(4.4%) | 42(13.2%) | 93(29.3%) | 149(15.7%) | |
| III | 72(22.7%) | 103(32.5%) | 30(9.5%) | 205(21.6%) | |
| IV | 52(16.4%) | 66(20.8%) | 4(1.3%) | 122(12.8%) | |
| Unknown | 103(32.5%) | 31(9.8%) | 1(0.3%) | 135(14.1%) | |
| **T category** | | | | | |
| T1 | 118(37.2%) | 89(28.1%) | 191(60.3%) | 398(41.9%) | <0.001 |
| T2 | 28(8.8%) | 57(18.0%) | 94(29.7%) | 179(18.8%) | |
| T3 | 37(11.7%) | 87(27.4%) | 27(8.5%) | 151(15.9%) | |
| T4 | 35(11.0%) | 47(14.8%) | 4(1.3%) | 86(9.0%) | |
| Tx | 99(31.3%) | 37(11.7%) | 1(0.2%) | 137(14.4%) | |
| **N category** | | | | | |
| N0 | 184(58.0%) | 223(70.3%) | 308(97.2%) | 715(75.2%) | <0.001 |
| N1 | 56(17.7%) | 70(22.1%) | 8(2.5%) | 134(14.1%) | |
| Nx | 77(24.3%) | 24(7.6%) | 1(0.3%) | 102(10.7%) | |

*(continued on next page)*

**Table 3** (*continued*)

| Characteristics | Histologic type | | | | |
|---|---|---|---|---|---|
| | Klatskin tumor | ICCA | HCC | Total | *p*-value |
| **Local treatment of the primary tumor** | | | | | |
| None | 293(92.4%) | 72(22.7%) | 20(6.3%) | 385(40.5%) | <0.001 |
| Local destruction | 0(0.0%) | 10(3.2%) | 40(12.6%) | 50(5.3%) | |
| Surgery | 24(7.6%) | 235(74.1%) | 257(81.1%) | 516(54.2%) | |
| **Alpha fetoprotein** | | | | | |
| Elevated | 22(6.9%) | 161(50.8%) | 316(99.7%) | 499(52.5%) | <0.001 |
| Normal | 62(19.6%) | 137(43.2%) | 1(0.3%) | 200(21.0%) | |
| Unknown | 233(73.5%) | 19(6.0%) | 0(0%) | 252(26.5%) | |
| **Fibrosis score** | | | | | |
| 0-4 | 14(4.4%) | 81(25.6%) | 229(72.2%) | 324(34.1%) | <0.001 |
| 5-6 | 2 (0.6%) | 24(7.6%) | 77(24.3%) | 103(10.8%) | |
| Unknown | 301(95.0%) | 212(66.8%) | 11(3.5%) | 524(55.1%) | |

**Notes.**

[a]Including divorced, separated, single (never married), unmarried or having a domestic partner and widowed.

[b]Including American Indian/Alaskan Native and Asian/Pacific Islander.

AJCC, American Joint Committee on Cancer; ICCA, Intrahepatic Cholangiocarcinoma; HCC, hepatocellular carcinoma.

congenital liver malformations (*Razumilava & Gores, 2014*); genetic abnormalities and molecular defects in various oncogenes may also play a role (*Okuda et al., 1977*).

The SEER database provided the population of patients for our research. However, there were limitations to our study. Serum CA19-9 expression is not widely used as a tumor marker in the SEER database (*Patel et al., 2000*) so we were not able to adjust for CA19-9 expression in the multivariate Cox analysis. Alpha fetoprotein (AFP) and hepatic fibrosis score status could only be separated into a single stratified study for prognosis. Due to the smaller sample size of only 317 Klatskin tumor patients enrolled, we were not able to fully calculate the mode. Differing treatments for liver cancer and cholangiocarcinoma also affected the prognosis. Additionally, the SEER database contains no data regarding chemotherapy, interventional therapy, or targeted therapy.

In conclusion, our study compared the clinical features of Klatskin tumors, ICCA, and HCC using Kaplan–Meier analysis and multivariate Cox analysis. We confirmed that the prognosis of Klatskin tumors was worse than that for ICCA or HCC. Age, lymph node status, summary stage, and the local treatment of the primary tumor were prognostic factors. Our nomogram objectively and accurately predicted the prognosis of Klatskin tumor patients.

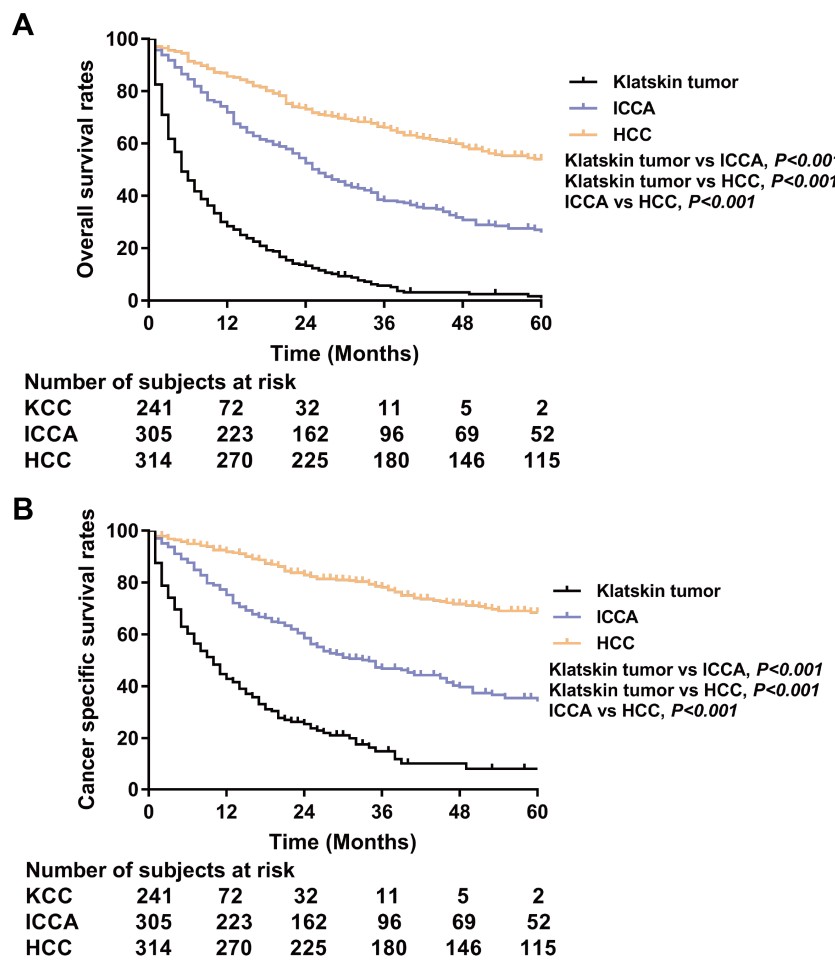

**Figure 4 Kaplan-Meier plot and log-rank test.** Overall survival (A) and cancer specific survival (B) according to histologic type of Klatskin tumors, ICCA or HCC after propensity score matching.

# ACKNOWLEDGEMENTS

We thank PeerJ for its linguistic assistance during the preparation of this manuscript.

## Funding

The authors received no funding for this work.

## Competing Interests

The authors declare there are no competing interests.

## Author Contributions

- Feng Qi conceived and designed the experiments, performed the experiments, analyzed the data, prepared figures and/or tables, and approved the final draft.

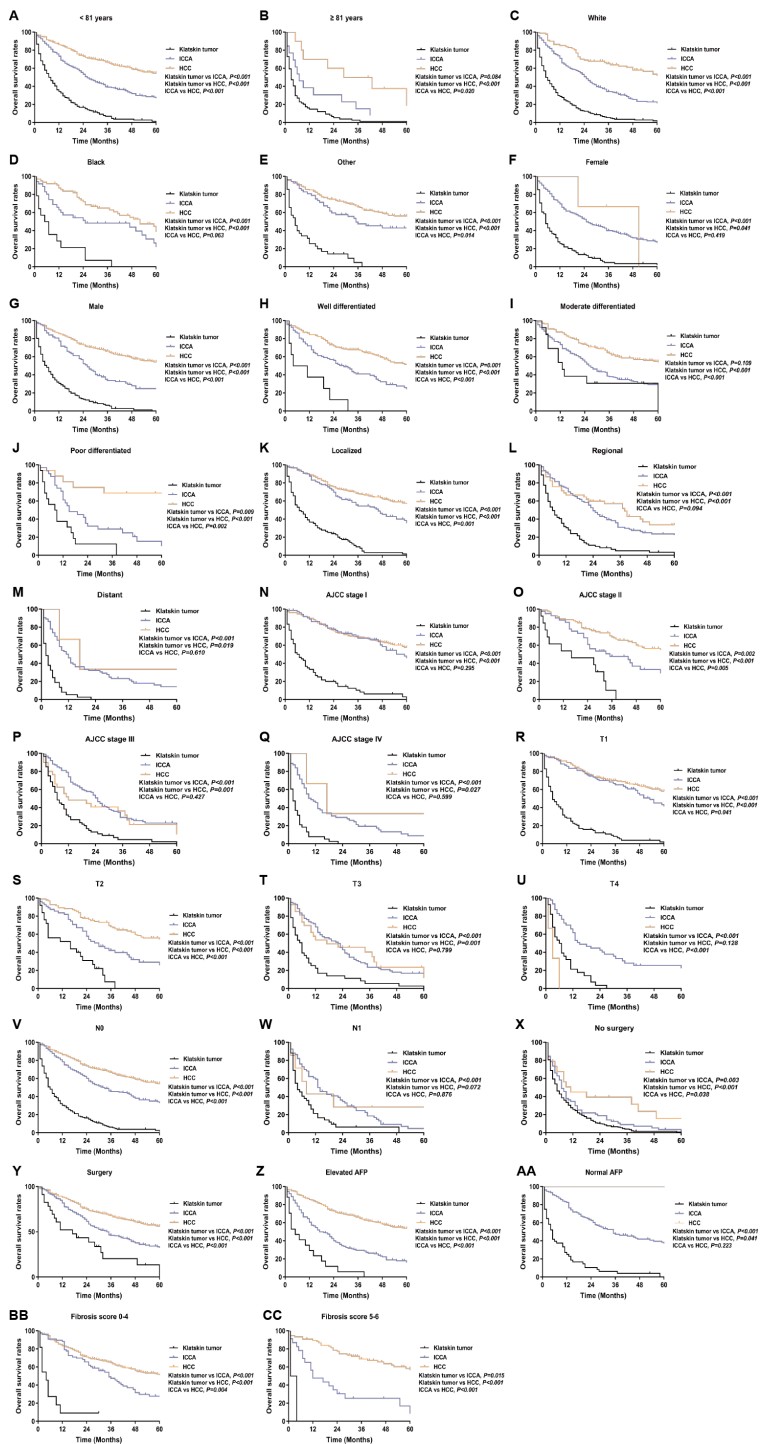

**Figure 5** Kaplan–Meier curve of overall survival in Klatskin tumor, ICCA and HCC patients after propensity score matching stratified by (A, B) age at diagnosis; (C-E) race; (F, G) gender; (H-J) primary tumor differential grade; (K-M) SEER historic stage; (N-Q) AJCC stage; (R-U) primary tumor size; (V, W) lymph node status; (X, Y) surgery for primary tumor; (Z, AA) AFP level; (BB, CC) fibrosis score.

**Table 4** Univariate and multivariate Cox proportional hazard analyses of the association of clinical characteristics with overall survival rates in patients with Klatskin tumors, ICCA and HCC after propensity score matching.

| Variance | Univariate HR (95% CI) | P value | Multivariate HR (95% CI) | P value |
|---|---|---|---|---|
| **Age** | | | | |
| <81 years | 1 | | 1 | |
| ≥81 years | 3.409(2.801–4.148) | <0.001 | 1.736(1.396–2.157) | <0.001 |
| **Race** | | | | |
| White | 1 | | 1 | |
| Black | 0.610(0.481–0.774) | <0.001 | 1.223(0.945–1.583) | 0.126 |
| Other | 0.503(0.420–0.604) | <0.001 | 0.976(0.796–1.198) | 0.819 |
| **Gender** | | | | |
| Female | 1 | | 1 | |
| Male | 0.679(0.585–0.789) | <0.001 | 1.057(0.878–1.271) | 0.559 |
| **Grade** | | | | |
| Well differentiated | 1 | | 1 | |
| Moderate | 1.029(0.843–1.257) | 0.778 | 0.872(0.691–1.100) | 0.249 |
| Poor | 1.470(1.072–2.016) | 0.017 | 0.966(0.669–1.396) | 0.854 |
| Anaplastic | 6.219(0.867–44.587) | 0.069 | 2.332(0.279–19.504) | 0.435 |
| **SEER historic stage** | | | | |
| Localized | 1 | | 1 | |
| Regional | 1.954(1.635–2.336) | <0.001 | 1.141(0.860–1.513) | 0.361 |
| Distant | 3.075(2.479–3.813) | <0.001 | 1.362(0.711–2.606) | 0.351 |
| **AJCC stage** | | | | |
| I | 1 | | 1 | |
| II | 0.862(0.672–1.105) | 0.241 | 1.273(0.720–2.250) | 0.407 |
| III | 2.085(1.700–2.557) | <0.001 | 1.376(0.875–2.163) | 0.167 |
| IV | 3.643(2.882–4.604) | <0.001 | 1.556(0.755–3.205) | 0.231 |
| **Stage T** | | | | |
| T1 | 1 | | 1 | |
| T2 | 0.855(0.686–1.067) | 0.166 | 0.787(0.472–1.314) | 0.360 |
| T3 | 1.754(1.416–2.172) | <0.001 | 1.060(0.730–1.539) | 0.760 |
| T4 | 2.181(1.688–2.818) | <0.001 | 1.041(0.695–1.560) | 0.844 |
| **Stage N** | | | | |
| N0 | 1 | | 1 | |
| N1 | 2.431(1.986–2.976) | <0.001 | 1.406(1.093–1.808) | 0.008 |
| **Local treatment of the primary tumor** | | | | |
| None | 1 | | 1 | |
| Local destruction | 0.233(0.166–0.328) | <0.001 | 0.635(0.421–0.956) | 0.030 |
| Surgery | 0.169(0.143–0.200) | <0.001 | 0.434(0.328–0.574) | <0.001 |
| **Alpha fetoprotein** | | | | |
| Elevated | 1 | | 1 | |
| Normal | 1.470(1.209–1.787) | <0.001 | 0.757(0.587–0.976) | 0.032 |
| **Fibrosis score** | | | | |
| 0-4 | 1 | | 1 | |

**Table 4** (*continued*)

| Variance | Univariate HR (95% CI) | P value | Multivariate HR (95% CI) | P value |
|---|---|---|---|---|
| 5-6 | 0.937(0.697–1.260) | 0.667 | 1.052(0.767–1.444) | 0.751 |
| **Histologic type** | | | | |
| Klatskin tumor | 1 | | 1 | |
| ICCA | 0.277(0.232–0.331) | <0.001 | 0.494(0.312–0.785) | 0.003 |
| HCC | 0.136(0.111–0.167) | <0.001 | 0.251(0.138–0.455) | <0.001 |

**Notes.**

CI, confidence interval; ICCA, Intrahepatic Cholangiocarcinoma; HCC, hepatocellular carcinoma.

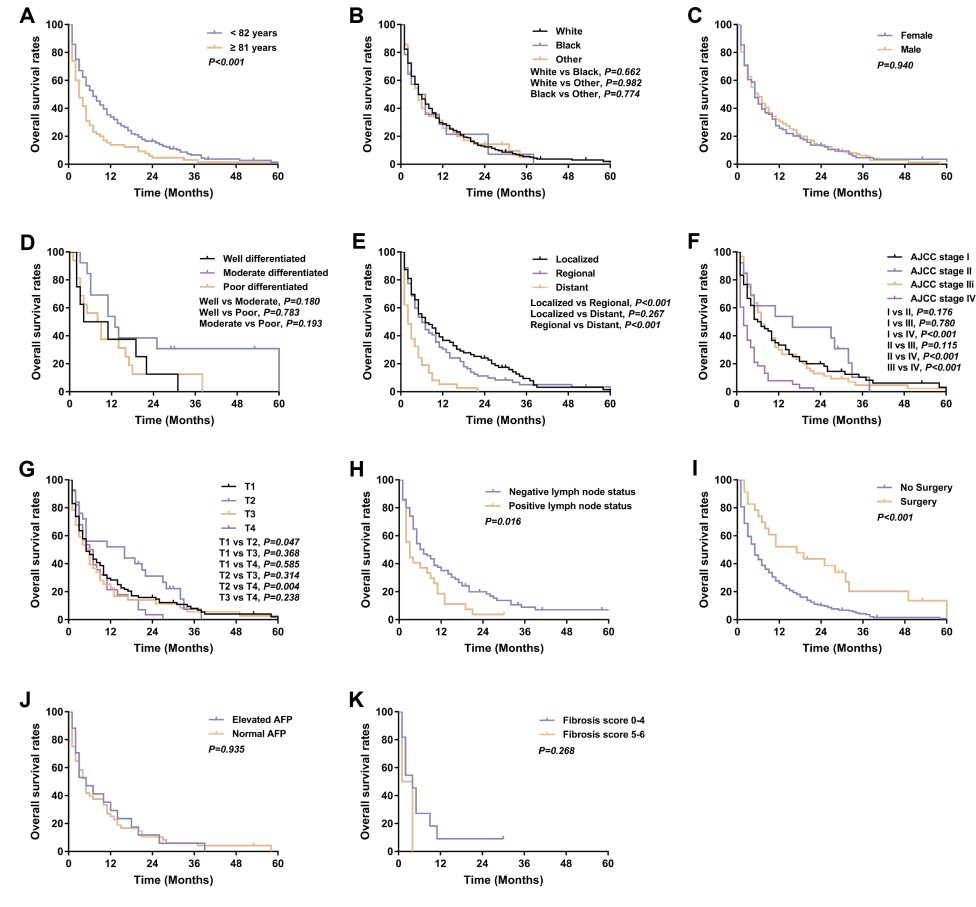

**Figure 6** Kaplan–Meier curve of overall survival in Klatskin tumor patients stratified by (A) age at diagnosis; (B) race; (C) gender; (D) primary tumor differential grade; (E) SEER historic stage; (F) AJCC stage; (G) primary tumor size; (H) lymph node status; (I) surgery for primary tumor; (J) AFP level; (K) fibrosis score.

- Bin Zhou analyzed the data, prepared figures and/or tables, and approved the final draft.
- Jinglin Xia performed the experiments, authored or reviewed drafts of the paper, and approved the final draft.

## Data Availability

**Table 5   Univariate and multivariate Cox proportional hazard analyses of the association of clinical characteristics with overall survival rates in patients with Klatskin tumors.**

| Variance | Univariate Hazard ratio (95% CI) | *P* value | Multivariate Hazard ratio (95% CI) | *P* value |
|---|---|---|---|---|
| **Age** | | | | |
| <82 years | 1 | | 1 | |
| ≥82 years | 1.663(1.305–2.121) | <0.001 | 1.725(1.324–2.249) | <0.001 |
| **Race** | | | | |
| White | 1 | | 1 | |
| Black | 1.098(0.696–1.734) | 0.687 | 1.735(1.063–2.831) | 0.027 |
| Other | 0.997(0.718–1.384) | 0.986 | 1.232(0.866–1.753) | 0.246 |
| **Gender** | | | | |
| Female | 1 | | 1 | |
| Male | 1.024(0.817–1.284) | 0.834 | 1.011(0.789–1.296) | 0.929 |
| **Grade** | | | | |
| Well differentiated | 1 | | 1 | |
| Moderate | 0.504(0.208–1.222) | 0.130 | 0.488(0.188–1.267) | 0.141 |
| Poor | 0.875(0.386–1.984) | 0.749 | 0.716(0.295–1.739) | 0.461 |
| Anaplastic | 1.335(0.169–10.570) | 0.784 | 3.911(0.370–41.325) | 0.257 |
| **SEER historic stage** | | | | |
| Localized | 1 | | 1 | |
| Regional | 1.168(0.878–1.553) | 0.287 | 1.878(1.144–3.082) | 0.013 |
| Distant | 2.164(1.521–3.079) | <0.001 | 3.594(1.251–10.326) | 0.018 |
| **AJCC stage** | | | | |
| I | 1 | | 1 | |
| II | 0.653(0.361–1.182) | 0.159 | 1.874(0.730–4.809) | 0.191 |
| III | 0.969(0.695–1.350) | 0.851 | 0.751(0.390–1.448) | 0.393 |
| IV | 2.020(1.401–2.914) | <0.001 | 0.723(0.248–2.106) | 0.552 |
| **Stage T** | | | | |
| T1 | 1 | | 1 | |
| T2 | 0.646(0.418–0.998) | 0.049 | 0.409(0.199–0.837) | 0.014 |
| T3 | 0.847(0.579–1.237) | 0.390 | 0.783(0.466–1.316) | 0.356 |
| T4 | 1.124(0.767–1.647) | 0.548 | 0.839(0.486–1.451) | 0.530 |
| **Stage N** | | | | |
| N0 | 1 | | 1 | |
| N1 | 1.301(0.957–1.768) | 0.093 | 1.468(1.008–2.139) | 0.046 |
| **Local treatment of the primary tumor** | | | | |
| None | 1 | | 1 | |
| Surgery | 0.393(0.246–0.630) | <0.001 | 0.555(0.316–0.977) | 0.041 |
| **Alpha fetoprotein** | | | | |
| Elevated | 1 | | 1 | |
| Normal | 1.015(0.622–1.655) | 0.953 | 1.077(0.643–1.805) | 0.778 |

**Table 5** (*continued*)

| Variance | Univariate Hazard ratio (95% CI) | *P* value | Multivariate Hazard ratio (95% CI) | *P* value |
|---|---|---|---|---|
| **Fibrosis score** | | | | |
| 0-4 | 1 | | 1 | |
| 5-6 | 1.419(0.320–6.299) | 0.645 | 0.682(0.141–3.302) | 0.635 |

**Notes.**
CI, confidence interval.

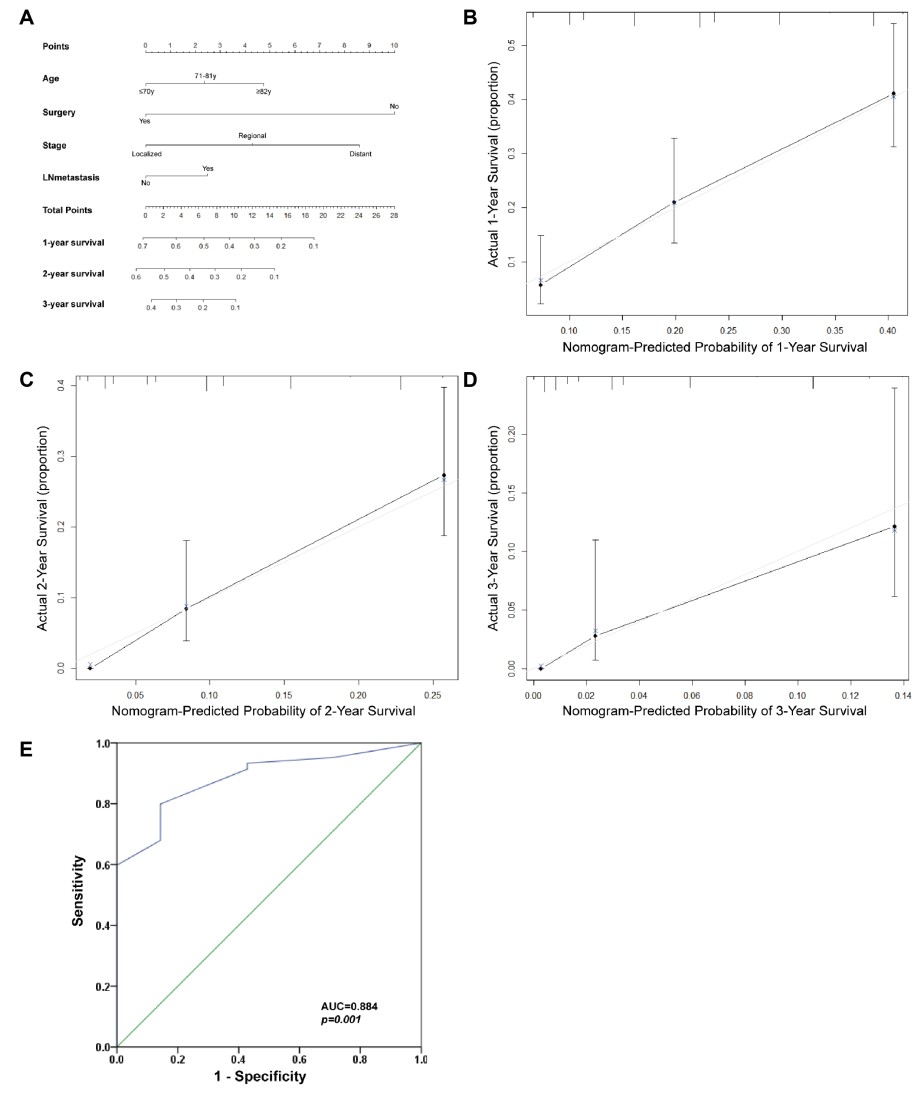

**Figure 7  Prognostic nomogram estimated by clinical characteristics for 1-year, 2-year, and 3-year OS in Klatskin tumor patients.** (A) To get the nomogram, the factors lie on each variable axis and is drawn up to determine the point value. A vertical line from the total point scale to the probability scale is draw and the probability of 1-year, 2-year, or 3-year OS is determined. The calibration curve for OS at (B) 1 years, (C) 2 years and (D) 3 years is shown. *X*-axis is nomogram-predicted OS and y-axis isactual OS. (E) ROC curve analysis for predicting the accuracy of the nomogram.

Data is available at Surveillance, Epidemiology and End Results (SEER) database, Primary Site-labeled: C22.0-Liver and C22.1-Intrahepatic bile duct from 2004 to 2014 and raw measurements are available in the Supplemental Files.

## Supplemental Information

Supplemental information for this article can be found online at http://dx.doi.org/10.7717/peerj.8570#supplemental-information.

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
