# Peer review of "Nomograms predict survival outcome of Klatskin tumors patients"

_PeerJ, doi:10.7717/peerj.8570_

## Round 0.1 · original submission · Major Revisions

Your manuscript has been reviewed and requires modifications prior to making a decision. The comments of the reviewers are included at the bottom of this letter. Reviewers indicated that the statistical methods section should be improved. Review 2 also indicated that the English language must be improved. I agree with the evaluation and I would, therefore, request for the manuscript to be revised accordingly. I would also like to suggest the following change:

Please recalculate the percentages of variables, because total percentages were less than 100% in some variables.

Reviewer 1 ·

Basic reporting

1. A flowchart of inclusion and exclusion criteria is required.

Experimental design

2. For analysis, “age” was converted from a continuous variant to a categorical variant as “< 50” and “≥ 50”. The authors need to explain why age is classified this way. Use software such as X-tile and R or refer to the NCCN guidelines to find the best cutoff value for the continuous variable.
3. In the statistical analysis, the author should note the format for the presented results, such as the mean ± S.E or median (range).
4. Continuous variables in normal distribution and homogeneity of variance were compared by using the t test, otherwise, the Mann-Whitney U test was performed.

Validity of the findings

5. A nomogram without external validation is only a visualization of Cox model. Therefore, the nomogram in this study should be placed after the results of Cox model.
6. In terms of model diagnosis, C-index alone is far from enough. First, the calibration curve for the nomogram is necessary. Besides, have you considered using some other penalization methods such as Elastic Net and Lasso regression to ensure the model not overfitting? One can look at the Cox-Snell residual plot, or Martingale residual plots as ways to assess model fit. ROC curve can evaluate the accuracy of the model.
7. The authors remove all patients with any kind of missing data. Maybe the authors want to consider using multiple imputation so they can include more patients. This way they could create two independent large datasets from the SEER data and use the small local dataset as a second independent validation set.

Reviewer 2 ·

Basic reporting

As described in general comments.

Experimental design

As described in general comments.

Validity of the findings

As described in general comments.

Additional comments

This is an interesting study by Qi et al on Klatskin tumors. Albeit its scientific significance, this study includes several major issues that should be addressed.
- The English language should be improved to ensure that an international audience can clearly understand the text.
- Why were patients only after 2005 included? Moreover, since the follow up period in the KM curves in 5y why patients until 2015 and not 2014 were included?
- Moreover, what was the follow up length and rate?
- Could the authors please specify their inclusion/exclusion criteria?
- Would the authors consider assessing the era effect?
- Could they provide more info regarding the treatment plan of the patients?
- Authors state that t-test was used, but there were 3 groups. How did they control for a error?
- In all Figures, authors are encouraged to provide patients at risk
- Could the authors provide a flowchart of the study showing total No of pts, reason for exclusion and No?
- There were significant differences among the 3 groups regarding patients’ demographic data. Have the authors considered performing propensity score matching analysis? Or a Nelson Aaalen cumulative hazard function including a multivariate analysis so that an adjusted risk can be calculated instead of a simple KM?
- Could the authors provide p-values between each 2 group comparison?

---

## Round 0.2 · accepted · Accept

The authors addressed the reviewers' concerns and substantially improved the content of MS. So, based on my own assessment as an editor, no further revisions are required and the MS can be accepted in its current form.

Reviewer 1 ·

Basic reporting

The manuscript can be accepted in this form

Experimental design

The manuscript can be accepted in this form

Validity of the findings

The manuscript can be accepted in this form

Additional comments

The manuscript can be accepted in this form

Reviewer 2 ·

Basic reporting

OK

Experimental design

OK

Validity of the findings

OK

Additional comments

OK to publish